# OpenReview forum: "HypoVeil: A Hypothesis-Driven Pragmatic Inference-Time Control Framework for Privacy–Utility-Aware LLM-Agent Dialogue"
_ICLR.cc/2026/Conference — ICLR 2026 Conference Withdrawn Submission_

### Official Review · Reviewer_ND7A · 2025-10-27

**Soundness:** 3
**Presentation:** 3
**Contribution:** 3
**Rating:** 4
**Confidence:** 2

**Summary:**

This paper proposes an inference-time mechanism to improve privacy preservation in conversational LLM agents.
The framework integrates (1) a hypothesis-driven mental model, maintaining natural-language “belief hypotheses” about interlocutors (their knowledge, motives, and likely interpretations), with (2) a Rational Speech Act (RSA)-based planner that selects utterances by maximizing task utility minus context-sensitive privacy cost.

**Strengths:**

1. The paper correctly isolates the under-addressed issue of inference-time privacy leakage—distinct from training-time memorization—and grounds it in contextual-integrity theory.
2. The combination of hypothesis-driven ToM reasoning with Rational Speech Act pragmatics for privacy control is original and theoretically elegant.
3. The six-dimension mental model (Knowledge, Request, Motive → Strategic Direction, Information Gaps, Privacy) is intuitively designed and operationally defined.

**Weaknesses:**

1. The method’s components—hypothesis tracking and RSA reasoning—each have prior art; the contribution is mainly integrative rather than algorithmically new.
2. Evaluation only on synthetic CI scenarios; no real-world human-LLM dialogue or deployment case, limiting ecological validity.
3. Both privacy and helpfulness metrics depend on automated judgments by the same or similar model family, risking circular evaluation bias.
4. Reported improvements (≈ 5 %) are small relative to the noise levels (± 3–4 %), raising questions about effect robustness beyond statistical significance.
5. Few concrete dialogue examples illustrating successful vs. failed privacy behavior; interpretability is claimed but not demonstrated.
6. V-BENCH scenarios, though CI-grounded, are generated by LLMs (GPT-4o) and thus may encode stylistic artifacts or training bias.

**Questions:**

-

---

> ### Author Response · Authors · 2025-12-03
>
> Thank you for these insightful and encouraging remarks. We are glad that our focus on inference-time privacy leakage, distinct from training-time memorization, was clear and well-motivated. We also appreciate your recognition of the conceptual contribution, here we address the concerns,
>
>
> > Clarification of Contribution
>
> We agree that both hypothesis tracking and RSA-style pragmatic reasoning have prior art, and we do not claim to introduce a new primitive inference algorithm. Our contribution is instead conceptual and integrative: to our knowledge, HYPOVEIL is the first framework that (i) maintains a dimension-aware hypothesis store explicitly targeting contextual privacy and (ii) couples this belief store with an RSA-style planner to optimize the privacy–utility in multi-turn dialogue. In this sense, HYPOVEIL proposes applied algorithms which is grounded in Theory-of-Mind style mental modeling and RSA pragmatics tailored specifically for contextual-integrity (CI) scenarios, while not aiming to provide new fundamental algorithmic primitives
> Prior RSA work does not operate over a multi-dimensional mental model that tracks what a counterpart knows, wants, and is likely to infer in CI scenarios. Our ablations show that neither component alone suffices: “mental w/o RSA” is overly conservative and harms utility, and “simple w/ RSA” fails to improve the trade-off, whereas “mental w/ RSA” consistently pushes the privacy–utility frontier outward across models. We therefore see HYPOVEIL’s main contribution as introducing and validating a privacy-aware ToM+RSA controller specialized for inference-time contextual privacy.
>
> > Concerns of evaluation only on synthetic CI scenarios
>
> We agree that evaluation on real-world human–LLM dialogues would further strengthen ecological validity; we see this as important future work. At the same time, the broader CI literature on LLMs (e.g., ConfAIde, PrivacyLens, CI-Bench, PrivaCI-Bench) has converged on synthetic or hybrid CI scenarios for precisely the reasons the reviewer hints at: collecting real conversations with rich private context is expensive, ethically sensitive, and often impossible to release. Our V-BENCH design follows this practice, but adds several dimensions that, to our knowledge, are not jointly present in prior CI benchmarks: a two-party multi-turn setting; explicit, graded sensitive inventories for Party A; explicit desired-information sets for Party B; calibrated overlap between the two; and strategic multi-turn probing with per-turn leak attribution as Table 4 in the paper summarizes
>
> > Concerns of circular evaluation bias
>
> We agree that using LLMs as judges can introduce risks of circularity if the same model is used for both conversation and evaluation. In our setup, the judge is a distinct, stronger closed-source model (GPT-4.1) that is not among the conversation models we evaluate (GPT-4o, Llama-3.1-8B, Gemma-3-27B). Evaluation prompts are rubric-based and tailored for judging, and we have explicit sensitive information and public information set. Moreover, we validate the judgment against human annotators: on a subsample of scenarios, three trained annotators compare pairs of model outputs under three axes (privacy protection, helpfulness, overall trade-off). We observe substantial agreement among humans and between humans and the LLM-judge: Light’s κ in the range 0.75 to 0.78.
>
> > Concerns of effect robustness beyond statistical significance.
>
> First, we present means with 95% CIs and scenario-blocked non-parametric tests (Friedman with Kendall’s W, with Holm-corrected post-hoc contrasts). These tests operate on per-scenario deltas and explicitly treat scenarios as blocks, which directly addresses the “noise versus effect” question. For example, on Llama-3.1-8B, Mental+RSA significantly outperforms Mental w/o RSA on the trade-off score and also reduces privacy risk. These effect sizes are small-to-medium but consistent across scenarios, as reflected in Kendall’s W in the 0.012–0.022 range. In contrast, experimental results for widely used method, CoT, are no better than or even worse than those for the baseline, aside from privacy risk for Gemma.
>
> > Need for More Concrete Dialogue Examples
>
> We appreciate the request for more illustrative cases; this aligns well with our goal of interpretability. The current version already includes a qualitative section in Appendix G with both scenario and conversation examples. For instance, Table 12 shows a representative case where Mental+RSA provides high-level behavioral advice (e.g., monitoring heuristics and generic time windows) while deliberately omitting patient-identifying medication names and dosages, whereas the Simple baseline reveals these sensitive details. We will provide additional examples in the camera-ready version to further demonstrate the methods’ behavior.

---

> > ### Author Response · Authors · 2025-12-03
> >
> > > Potential Bias in LLM-Generated CI-Grounded V-BENCH Scenarios
> >
> > Our multi-turn dialogues are generated by GPT-4o, and we acknowledge this as a limitation in terms of possible stylistic artifacts. However, recent CI benchmarks also rely on synthetic generation, and our design follows their works. V-BENCH is grounded in human-authored CI seeds from PrivacyLens, these seeds are then expanded through a multi-stage generate and verify pipeline with strict norm checks to ensure the scenarios and seeds follow human social norms.

---

### Official Review · Reviewer_5JEp · 2025-11-01

**Soundness:** 3
**Presentation:** 2
**Contribution:** 2
**Rating:** 4
**Confidence:** 3

**Summary:**

This paper proposes HypoVeil, an inference-time framework that enhances LLM agents’ privacy–utility trade-off by combining a hypothesis-driven mental model with a Rational Speech Act planner. The system explicitly models a conversation partner’s knowledge and motives to decide what to reveal or conceal. A new benchmark, V-BENCH, evaluates privacy-sensitive multi-turn dialogues. Experiments on GPT-4o, Llama-3.1-8B, and Gemma-3-27B show that HypoVeil improves privacy and helpfulness over strong baselines, demonstrating a principled approach to privacy-aware reasoning for LLM agents.

**Strengths:**

1. The paper examines LLM privacy through the lens of Contextual Integrity (CI), a newly-presented and well-suited framework for analyzing privacy dynamics in modern agent-based dialogues.

**Weaknesses:**

1. My main concern is how much contributions and insights that the paper can bring here. There has been extensive work on LLM privacy. While I am not familiar with prior work on CI, does all these previous methods cannot be evaluated under this CI settings? What benefits can HypoVeil offer compared to existing approaches in similar directions?
2. It will be much better to have a pseudo-code for HypoVeil to help readers understand all the steps listed from Section 3.
3. From my understanding, the core idea of HypoVeil is to generate multiple candidate utterances and select the one that minimizes privacy leakage. Similar ideas have been explored in other line of work of LLM privacy. For instance, the Exponential Mechanism, a standard approach in differential privacy, selects an optimal output from a set of options based on scores computed from sensitive data. This mechanism has inspired several prior studies [1, 2, 3] on privacy preservation in LLMs.

[1] Amin, Kareem, et al. "Private prediction for large-scale synthetic text generation." EMNLP 2024.

[2] Flemings, James, Meisam Razaviyayn, and Murali Annavaram. "Differentially private next-token prediction of large language models." ACL 2024.

[3] Bhusal, Bishnu, et al. "Privacy-Aware In-Context Learning for Large Language Models." ICLR 2024.

**Questions:**

1. Is it possible to extend this setting (Party A v.s. Party B) to a multi-party (>2) conversation? It will fit more in modern multi-agent systems.
2. How is the Leak score calculated in Eq. 2? It is stated that it comes from the Privacy/Sensitivity dimension. How are these values calculated originally in Sec. 3.1?

---

> ### Author Response · Authors · 2025-12-03
>
> Thank you for recognizing our use of Contextual Integrity as a suitable framework for analyzing privacy in modern agent-based dialogue. Here we address the concerns,
>
> > Contributions and insights that the paper can bring
>
> We appreciate the request for a clearer comparison to recent contextual-privacy benchmarks. Technically, V-BENCH is designed to complement, rather than duplicate, frameworks such as ConfAIde, PrivacyLens, and MAGPIE. PrivacyLens instantiates contextual-integrity seeds in a single-agent, tool-use setting, where an LM agent executes a user instruction via tools and is evaluated on whether the final action leaks sensitive items extracted from the trajectory. MAGPIE moves to multi-agent, multi-turn collaboration, but in non-adversarial scenarios where private information is task-critical; it does not expose an explicit sensitive-information inventory for the disclosing party nor a structured desired-information set for the counterpart to test utility-privacy tradeoff. In contrast, V-BENCH is explicitly constructed as two-party, multi-turn dialogue where (i) Party A is equipped with an itemized sensitive-information inventory annotated on a 0–5 scale, (ii) Party B has an explicit desired-information set tied to its task goal, and (iii) the overlap between these two sets is calibrated to create real privacy–utility pressure. We further script Party B’s prompts to strategically escalate information requests over turns, and we perform per-turn leak attribution and LLM-as-Judge scoring for both helpfulness and privacy. Appendix C now includes Table 4, which summarizes these dimensions and highlights that no prior CI benchmark jointly supports multi-sensitive information, explicit desired-sets with calibrated overlap, multi-turn strategic probing, and per-turn leak attribution under a CI frame; this specific combination is precisely what HypoVeil requires to study inference-time privacy reasoning.
>
> We have expanded the related-work discussion to more clearly position HypoVeil relative to other CI methods such as AirGapAgent and Firewalls. Both lines of work focus on system-level defenses that restrict what context an agent can access (e.g., air-gapping tools, constructing task-specific firewalls against indirect prompt injections). HypoVeil studies inference-time dialogue planning: the agent maintains a hypothesis-driven mental model over what the partner knows and seeks, then uses an RSA-style pragmatic objective that explicitly trades off utility against context-sensitive privacy cost in multi-turn conversation. We clarify in the revised version that existing defenses could in principle be evaluated on V-BENCH by wrapping their policies around our Party-A agent, but their released implementations and threat models are currently tailored to different environments (tool hijacking, indirect prompt injection, etc.), which is why we focus our empirical study on baselines (Simple / CoT and ablations of the mental model and RSA) that share the same multi-agent conversational setting.
>
>
> > Pseudo-code
>
> In Appendix A, we provide concrete input/output specifications and operational definitions for every subroutine. All hyperparameters, FAISS retrieval behavior, confidence calibration, and RSA scoring details are now fully documented in Appendix A.

---

> ### Author Response · Authors · 2025-12-03
>
> > The core idea of Hypoveil
>
> We appreciate the reviewer’s observation, but HypoVeil is fundamentally different from approaches that simply generate multiple candidate utterances and select the one with minimal privacy leakage. Prior work typically treats privacy as a static filtering problem and evaluates disclosure based on fixed sensitive attributes. In contrast, our setting is grounded in Contextual Integrity (CI), where the permissibility of sharing information depends on the dynamic relationship between the sender and recipient, the evolving conversational context, and the interlocutor’s inferred intent. As we show in Table 4 (newly added) and in the V-BENCH design, our benchmark explicitly constructs multi-turn scenarios with calibrated overlap between Party B’s goals and Party A’s sensitive attributes, together with public information that should still be shared for utility. This makes privacy preservation a context-dependent reasoning challenge rather than a static filtering task following CI setting.
>
> Within this setup, HypoVeil provides a solution for CI-governed multi-turn dialogue. The hypothesis-driven mental model explicitly tracks what the partner likely knows, wants, and would infer, while the RSA module reasons pragmatically about how different utterances advance goals under these beliefs. This coupling is crucial: as shown in Table 2 and Table 3, using RSA alone (i.e., generating candidates and selecting the least risky option) does not outperform a naive LLM, whereas combining RSA with the hypothesis store meaningfully improves both privacy and utility. Our ablation studies demonstrate that the gains arise from hypothesis formation, belief tracking, and pragmatic inference, not from candidate enumeration alone. Thus, HypoVeil contributes a novel integration of hypothesis-driven ToM reasoning with RSA-based evaluation to realize inference-time control that aligns with contextual integrity, a direction not explored by the prior methods.
>
> > Extension to Multi-Party (>2) Conversational Settings
>
> We agree that extending beyond a two-party setting is both feasible and valuable for modern multi-agent systems. In principle, our hypothesis-driven framework is well suited for multi-party interaction, since the mental model is explicitly designed to analyze and track the intentions, knowledge states, and likely interpretations of communication partners, and to internally consolidate this reasoning before selecting the most appropriate response. This capability naturally generalizes to settings where more than two agents interact, collaborate, or negotiate. However, due to the inherent complexity of scenario design in contextual-integrity–grounded privacy tasks, where roles, transmission norms, and information-overlap structures must be precisely controlled, we focus on two parties in this work to establish a clear and auditable evaluation protocol. We agree that extending hypothesis-driven methods to multi-party communication and negotiation is a highly promising direction, and we see this as an important avenue for future work. We will add a limitation section in the camera-ready version and describe this future direction.
>
> > Clarification on Leak Score Computation
>
> We thank the reviewer for the question. Since our goal is to evaluate and improve the model’s own ability to align with contextual privacy norms, the leak score used in Eq. 2 is derived implicitly from the model itself. Specifically, the Privacy/Sensitivity dimension in Sec. 3.1 assigns a sensitivity level based on the model’s internal hypothesis updates, and the leak penalty is then computed by an internal self-LLM judge that evaluates whether a candidate utterance reveals content that the mental model considers sensitive. In the revised version, we have clarified this design choice in Section 3.1 and emphasized that the scoring mechanism relies on the model’s own privacy-aware reasoning, similar to a self-refinement step in which the model provides its own normative assessment. As demonstrated by our ablation studies, this step is critical: Table 1 and Figure 2 show the comparison between mental w/o RSA and mental w/ RSA, indicating that incorporating this LLM-derived leak assessment improves the privacy–utility balance.

---

### Official Review · Reviewer_J1KV · 2025-11-01

**Soundness:** 3
**Presentation:** 3
**Contribution:** 3
**Rating:** 6
**Confidence:** 4

**Summary:**

The paper proposes to use hypothesis framework to manage what an agent should share or not with the counterpart. It relies on six-dimension model and introduces a benchmark with many scenarios to test it. The paper provides a nice outline for future agentic development and deployment of AI Assistants with access to user information.

**Strengths:**

- Paper addresses a key problem in privacy for AI Agents --> how can we know what information can be shared or not
- The paper adds hypothesis and RSA view of the communication
- Paper also introduces the benchmark for multiple scenarios to test multi-turn privacy
- experiments demonstrate strong improvements over baseline methods

**Weaknesses:**

- It is not clear how this method will work under attacks that attempt to steal data from the agent. Similar to prompt injections [1], [2] and [3] show that agents are easily persuaded to share information. It might be great to have a discussion.
- Furthermore, given the proposed hypothesis method the adversary can attempt to manipulate the the hypothesis by the model to force it to increase trust and share more information
- It is possible to isolate the agent before interactions by minimizing the data [2], however it might require assuming some hypothesis before interactions. It might be great to have a discussion on these tradeoffs.


[1] Agentdam: Privacy leakage evaluation for autonomous web agents, Arxiv'25
[2] AirGapAgent: Protecting Privacy-Conscious Conversational Agents, CCS'24
[3] The Sum Leaks More Than Its Parts: Compositional Privacy Risks and Mitigations in Multi-Agent Collaboration, Arxiv'25

**Questions:**

addressing threat model questions might significantly strengthen the paper.

---

> ### Author Response · Authors · 2025-12-03
>
> Thank you for your thoughtful and supportive comments. We appreciate your recognition of the importance of the privacy problem in AI agents and are glad that our integration of hypothesis tracking and RSA reasoning resonated with you. Here we address the main concerns
>
> > How this method will work under attacks
>
> We thank the reviewer for raising the important issue of how HYPOVEIL behaves under active data-extraction attacks. Prior work such as Agentdam and AirGapAgent demonstrates that LLM agents can be persuaded (via prompt injection, social-engineering prompts, or compositional attacks) to reveal information they would otherwise protect. We agree that discussing this threat model is valuable for strengthening the paper. Conceptually, HYPOVEIL is designed to provide inference-time protection even when the interlocutor is adversarial, because the privacy constraint is anchored in Party A’s fixed sensitive-information inventory and the model’s internally maintained Privacy/Sensitivity dimension. Thus, even if an attacker attempts to coerce, mislead, or escalate the dialogue, the RSA evaluation layer continues to penalize candidate utterances that would disclose sensitive attributes, and the privacy boundary is never relaxed by the attacker’s apparent cooperation. As a result, adversarial escalation may change the perceived “strategic direction” of the Party B’s conversation attitude in the mental model but cannot override the explicit privacy guardrails encoded in the mental model and the RSA scoring. We add more description in Section 3.1.
>
> > Force it to increase trust and share more information
>
> We appreciate the reviewer’s observation regarding the possibility of an adversarial Party B attempting to manipulate the hypothesis store. Our design explicitly guards against this threat. Even if Party B strategically adjusts their utterances to appear more cooperative or trustworthy, the hypotheses generated by HYPOVEIL never override the foundational privacy commitments of Party A. The mental model operates under a fixed sensitive-information inventory and a persistent Privacy/Sensitivity dimension, meaning that all downstream reasoning remains anchored to Party A’s private constraints rather than Party B’s framing. In other words, while a more accommodating or strategically crafted response from Party B may lead the system to generate more helpful or task-relevant hypotheses, it cannot cause the agent to relax its assessment of what information is permissible to disclose. We add more description in Section 3.1.
>
> > Isolate the agent before interactions by minimizing the data
>
> This might need more ablation study on the “can be accessed data”, we think this is out of the scope of our paper but it is worth discussing. Our concern is that our evaluation is conducted on a multi-turn dataset where the benchmark contains overlapping information between Party B’s goal and Party A’s privacy sensitive content, along with public information that can be safely communicated. We aim to measure whether the model can autonomously determine which information should be protected in the current context. In this setting, if we force the model to begin with a minimal version of the story background which includes both privacy related and public information, though the private details are never explicitly revealed (see Table 10 for specifics). This will become difficult to control what this minimalization should look like. This creates an additional layer of filtering over privacy versus public information. However, according to contextual integrity, the permissibility of information transmission depends on the specific situation and the roles of the communicating parties. Therefore, such filtering would likely require an additional agent that removes private information from the story, which partly overlaps with what our hypothesis track already evaluates, namely whether the agent can correctly identify privacy sensitive content. Incorrect filtering could accidentally remove public information, thereby harming utility.

---

### Official Review · Reviewer_wqAs · 2025-11-01

**Soundness:** 3
**Presentation:** 1
**Contribution:** 2
**Rating:** 4
**Confidence:** 4

**Summary:**

The paper proposes **HypoVeil**, an inference-time privacy framework that combines hypothesis-driven reasoning with pragmatic decision-making. By maintaining a belief store of hypotheses about the interlocutor’s knowledge and intentions, and integrating a Rational Speech Act (RSA) module to optimize utility–privacy trade-offs, the method enables agents to act contextually while preserving privacy. Its effectiveness is demonstrated on **V-Bench**, a benchmark designed for multi-turn privacy-sensitive interactions.

**Strengths:**

1. The evaluations are well designed, effectively accounting for the trade-off between privacy and helpfulness.
2. The proposed method operates entirely at inference time, requiring no additional training, and can be readily applied to both small models (e.g., Llama 3.1) and large ones (e.g., GPT-4o).
3. Research on enhancing inference-time reasoning through contextual integrity is of great importance, and I believe this paper is strongly motivated in that direction.

**Weaknesses:**

1. Clarity and Motivation

This paper is generally difficult to grasp in terms of its overall idea, motivation, and methodology. For example, *hypothesis* seems to play a central role according to the title, yet Figure 1 (the overview) does not reflect this concept at all. Moreover, the Introduction does not sufficiently explain what *hypotheses* are proposed in this paper or how they guide the reasoning process of LLM-agents. I noticed a brief mention of this in Line 150 (Section 2), but it is far from adequate. Since the main innovation of the paper appears to lie in its hypothesis-driven design, the authors should provide a clearer and more detailed explanation—possibly supported by concrete data examples or illustrative figures.

Additionally, several key terms are not well defined. For instance, the notions of *evidence* in Section 3 are hard to interpret. The word *evidence* first appears in Line 168, but its meaning and role are unclear—why would multi-turn dialogue require *evidence*? I assume these terms (along with others such as *utterance*) may originate from social science theories, but since most of the target audience is likely unfamiliar with that background, it would be helpful to include explicit definitions or explanations.

2. Experiments

The experiments are not sufficiently extensive. From what I can tell, Figures 2 and Tables 1–3 all appear to be derived from the same experiment runs. And the conclusions drawn from them are not particularly informative.

3. Methodology

I cannot clearly identify the novelty of the proposed method, as it is not sufficiently explained how the introduced hypotheses contribute to improving reasoning performance.

**Questions:**

Please see weaknesses.

---

> ### Author Response · Authors · 2025-12-03
>
> Thank you for highlighting these strengths. We are glad that you found the evaluation design effective in capturing the privacy–helpfulness trade-off and that the inference-time nature and broad applicability of our method came through clearly. Here we address the concerns
>
> > Concern of the clarity and motivation
>
> We thank the reviewer for pointing out the need for clearer conceptual grounding of hypotheses and evidence. In the revised version, we clarify that a hypothesis in HYPOVEIL refers to a concise natural-language belief about the interlocutor or the evolving conversational state, maintained across six dimensions (Knowledge/Expertise, Request/Behavior, Motive/Trust, Strategic Direction, Information Gaps, and Privacy/Sensitivity). As described in Sec. 3.1, each hypothesis takes the form $ (h_i, E_i, c_i) $, where $ h_i $ is a one-sentence belief, $ E_i $ is a set of quoted evidence spans from the transcript or retrieved artifacts, and $ c_i $ is a calibrated confidence score. These hypotheses serve as an explicit inference-time reasoning substrate: they track what Party B likely knows, seeks, or intends, and also guide Party A’s own strategic options (e.g., partial disclosure, deferral, or clarification). Figure 1 now reflects this pipeline by showing hypotheses as the structured “belief store” that feeds into utterance planning.
>
> We further clarify the role of evidence. Because hypotheses evolve over a multi-turn exchange, the model must justify updates with specific spans from the conversation or retrieved context. The evidence lists $E_i$ enable HYPOVEIL to maintain transparency and coherence across turns and support later consolidation or paraphrase operations (Section 3.1). As described in the retrieval backend, evidence snippets are embedded, stored in a FAISS embedding storage, and used to retrieve semantically adjacent priors when new messages arrive. This design is helpful for multi-turn reasoning: without explicit evidence, the model would struggle to maintain consistent beliefs, perform merge-versus-create updates, or provide the calibrated confidence scores $ c_i $ that guide RSA-based planning. Hypotheses supply the structured beliefs that govern privacy-aware reasoning, while evidence provides the traceable grounding that keeps these beliefs coherent and interpretable over the full dialogue.
>
> > Concern of the experiments are not sufficiently extensive
>
> We appreciate the reviewer’s concern. Our experiments are indeed built around a unified ablation framework, but they are far from a single analysis. Figures 2 and Tables 1–3 reflect three complementary experimental views of the same controlled evaluation: (i) aggregate performance across all scenarios, (ii) Pareto-frontier comparisons showing the privacy–utility trade-off, and (iii) paired-scenario statistical significance tests using blocked-design Friedman analyses with Holm correction. These analyses are intentionally derived from the same experimental runs so that the effects of each ablation are directly comparable. The ablation suite is extensive: it tests the full model, hypothesis-only, RSA-only, simple baselines, and single-agent vs two-agent settings across 166 calibrated multi-turn scenarios. All three reporting formats reveal the same consistent pattern: the mental model alone is overly privacy-protective and loses utility, while the integration of RSA with the hypothesis track restores balance and yields the dominant privacy–utility frontier. The convergence of results across multiple metrics and statistical tests supports the robustness of our findings.
>
> > Concern of the novelty of the proposed method
>
> We thank the reviewer for highlighting the need for clearer explanation. To the best of our knowledge, our work is the first to integrate an explicit hypothesis track which stores the model’s internal beliefs about the partner’s knowledge, intent, and potential inferences, with an RSA (Rational Speech Acts) pragmatic selection mechanism for inference-time privacy control. The hypothesis track alone does not simply improve helpfulness; in fact, Table 3 shows the opposite: its privacy dimension becomes overly conservative, reducing leakage but significantly harming utility. This behavior validates our design assumption that hypotheses capture rich “mental-model” structure but require a pragmatic decision layer to be used effectively. The novelty of our method therefore lies in this combination: hypotheses provide structured, multi-dimensional internal reasoning, and RSA aligns these internal beliefs with externally pragmatic, contextually appropriate responses, yielding balanced privacy–utility behavior. Our ablations demonstrate that neither component by itself is sufficient, but the integration produces significant improvements, empirically confirming the contribution of the mental-model hypotheses to reasoning performance and contextual privacy control.

---

### Official Review · Reviewer_XNdJ · 2025-11-11

**Soundness:** 1
**Presentation:** 1
**Contribution:** 2
**Rating:** 2
**Confidence:** 3

**Summary:**

The paper considers the setup of two LLM agents interacting with each other, and focuses on the problem that an LLM agent may 'overshare' information that is not contextually relevant, thereby resulting in privacy violation. Similar to some recent works, the paper studies this problem of 'unintended' disclosures in LLM agents from the perspective of contextual integrity [Nissenbaum 2004]. The paper proposes an inference-time privacy method called HypoVeil. Each party maintains a 'mental model' in terms of one-sentence hypothesis, list of embeddings of conversation history, and calibrated confidence scores. This is used for planning the response in upcoming turns.  The proposed method also leverages so-called Rational Speech Act (RSA). The paper also proposes a benchmark V-Bench for evaluating utility versus privacy in LLM agent interactions.

**Strengths:**

* The problem that LLM agents may disclose unintended information during interactions is practically important, and has recently started to get attention. This makes the paper timely.
* The paper leverages ideas from contextual integrity, which provides an interesting platform for designing solutions to handle the privacy problem.

**Weaknesses:**

* There are two big concerns. First, the paper does not put the contributions into the context with respect to recent works on privacy of interactions between LLM agents.
    - Comparisons with prior contextual privacy benchmarks, especially PrivacyLens [Shao et al., 2024] and Magpie [Juneja et al., 2025] are cursory and vague. Lack of comparison in terms of technical aspects makes it difficult to asses the novelty of contributions of the V-Bench benchmark. The paper also does not include experimental results of prior benchmarks such as PrivacyLens. It would have been helpful to show what leakage aspects (or scenarios with specific technical examples) are covered by V-Bench that PrivacyLens or Magpie fail to capture. See Questions section for details
    - The paper does not consider some of the relevant works such as AirgapAgent [Bagdasarian et al., CCS 2024] and Firewalls [Abdelnabi et al., 2025].
* Second, several concepts in the proposed method of HypoVeil lack technical details, thereby limiting the technical rigor of the paper. For instance, many sub-routines in Algorithm 1 and 2 are not defined in detail. This makes understanding the technical details tricky and would also make reproducibility challenging.

References:
1. Bagdasarian et al., "AirGapAgent: Protecting Privacy-Conscious Conversational Agents", SIGSAC Conference on Computer and Communications Security (CCS), 2024
2. Abdelnabi et al., "Firewalls to Secure Dynamic LLM Agentic Networks", 2025

**Questions:**

- Can the authors expand on technical differences between V-Bench and prior benchmarks PrivacyLens [Shao et al., 2024] and Magpie [Juneja et al., 2025]? The related works section mentions that "However, they leave open key needs for inference-time study: a testbed that stresses multi-turn, calibrated overlap to create privacy protection pressure, strategic probing than a single response. Our V-Bench addresses these gaps by ...". However, this description is vague and does not capture technical differences. The paper claims to make CI tuples explicit, but PrivacyLens also has CI tuples, and the paper indeed samples CI seeds from PrivacyLens. It is important to make the differences more explicit with technical/conceptual rigor.
- Several questions remain open: Is the only difference of V-Bench due to multi-turn consideration? Are there explicit scenarios where prior benchmarks fail to capture privacy leakage? Are there experiments showing numerical comparisons with prior benchmarks?
- How does HypoVeil compare against prior contextual privacy baselines such as AirgapAgent and Firewalls?
- What is the difference between simple and chain-of-thought baselines? Does simple mean use LLM agent as-is and does CoT simply use a different prompt? It will be helpful to add more details.
- For GPT-4o and  Llama3-8b, "mental w/ rsa" achieves higher helpfulness than simple and CoT baselines. (In Gemma3-27b, helpfulness scores are very close.) This seems counterintuitive. Simple baselines without privacy requirements often achieve higher helpfulness at the cost of privacy. Can the authors provide details on how "mental w/ rsa" improves helpfulness while also reducing privacy leakage? Explicit examples showing qualitative improvement will be very helpful.

---

> ### Author Response · Authors · 2025-12-03
>
> Thank you for your feedback. We appreciate your recognition of the importance and timeliness of the privacy challenge in LLM agent interactions, here we address the concerns:
>
> > Positioning V-BENCH relative to prior contextual-privacy benchmarks
>
> We appreciate the request for a more precise technical comparison to PrivacyLens and MAGPIE. From a high level, V-BENCH follows the same design as recent CI benchmarks which are LLM-generated scenarios grounded in contextual integrity, but it targets a different evaluation need: inference-time multi-turn privacy reasoning under calibrated overlap between what one party wants and what the other party must keep private. Concretely, the revision now adds a dedicated comparison Table 4 in Appendix C that contrasts V-BENCH with PrivacyLens and MAGPIE along multiple axes, including (i) single-agent vs multi-agent setup, (ii) presence of an explicit graded sensitive-information inventory for the disclosing party, (iii) presence of an explicit desired-information set for the counterpart, (iv) whether overlap between these sets is calibrated and scenario-controlled.Technically, PrivacyLens instantiates CI tuples in a single-agent, tool-use setting where the model executes a task via tools and is evaluated on whether the final trajectory leaks any sensitive items extracted from the seed; it does not explicitly construct a multi-agent dialogue where one agent’s desired set overlaps with the other’s sensitive inventory, nor does it measure per-turn leakage during a strategically probing conversation. MAGPIE, in turn, considers multi-agent collaboration focuses on non-adversarial scenarios where private information is often task-critical and is not paired with an explicit desired-set vs sensitive-set overlap that induces privacy-utility pressure at each turn. The Party B attitude shifted from harmless small talk to increasingly targeted and context-appropriate probes that subtly pressure Party A toward revealing high-overlap items without appearing overtly adversarial. Evaluation then tracks per-turn leak decisions and helpfulness under this controlled pressure.
>
> Additionally, regarding CI tuples, we acknowledge that existing frameworks already include them. This is a prerequisite for any CI benchmark [1], and therefore not the novelty of our work. Our use of CI tuples simply follows what CI benchmarks are expected to provide in terms of explicitly displaying these conditions.
>
>
> > Algorithm not defined in detailed
>
> We appreciate the request for greater clarity. In the revision, we substantially expanded Algorithms 1 and 2 and added a new “Algorithmic Components” section in Appendix A that provides concrete input/output specifications and operational definitions for every subroutine. All hyperparameters, FAISS retrieval behavior, confidence calibration, and RSA scoring details are now documented in Appendix A.
>
> > HypoVeil compare against prior contextual privacy baselines
>
> We thank the reviewer for pointing out these relevant lines of work. In the revision, we have expanded the related-work section to clarify how HYPOVEIL relates to system-level defenses such as AirGapAgent and Firewalls. Both AirGapAgent and Firewalls primarily operate by constraining what context the agent can access such as air-gapping tools, limiting reachable resources, or constructing firewalls against indirect prompt injection and compositional attacks. HYPOVEIL is in a different setting: we focus on inference-time dialogue planning given that context. Our agent maintains a hypothesis-driven mental model of what the counterpart likely knows and wants, and uses an RSA-style pragmatic objective that explicitly trades off task utility against privacy cost in multi-turn conversation. Because AirGapAgent and Firewalls are designed and evaluated in different environments (e.g., web-browsing agents, tool-calling workflows) and their implementations are not directly aligned with our two-party conversational setting, a controlled empirical comparison within V-BENCH is not straightforward for this submission.

---

> ### Author Response · Authors · 2025-12-03
>
> > More details about simple and cot baseline
>
> We agree that the definitions of the Simple and CoT baselines should be clearer. We provide more details in the revised version in Appendix D, we follow this paper [2] to design our chain-of-thought prompt, here are the difference:
>
> Simple: a single LLM agent prompted with high-level instructions to be helpful while avoiding unnecessary disclosure, and given access to the full conversation history. No explicit hypothesis store or RSA planner is used; the model responds directly in one shot per turn.
>
> CoT: the same agent and instructions as Simple, but now prompted to let the model think about Party B’s intent and the privacy situation before producing a final answer. The reasoning scratchpad is used only within the model (not exposed to the counterpart) and no explicit multi-dimensional hypothesis store or RSA scoring is applied.
> Thus, CoT differs from Simple only in prompt structure and the presence of a free-form reasoning channel.
>
> > More details on how "mental w/ rsa" improves helpfulness while also reducing privacy leakage
>
> Both the mental and simple model prompts let them think about the privacy requirements. But the difference is the mental model reasoning Party B’s intent, storing their intent and reasoning how their requirements will cost privacy leakage. Because the mental model with RSA design is explicitly introduced to improve both utility and privacy protection, the agent internally performs deeper reasoning to infer Party B’s intent and respond in a more contextually appropriate way. After generating a candidate response, the RSA component further performs a two-step evaluation, assessing both the utility of the reply and whether it appropriately manages sensitive information. Altogether, this leads to consistent improvements in both utility and privacy outcomes.
>
> [1] Shvartzshnaider, Yan, and Vasisht Duddu. "Position: Contextual Integrity is Inadequately Applied to Language Models." arXiv preprint arXiv 2025.
>
> [2] Lan, Guangchen, et al. "Contextual integrity in LLMs via reasoning and reinforcement learning." arXiv preprint arXiv 2025.

---

### Note · Authors · 2026-01-07

I have read and agree with the venue's withdrawal policy on behalf of myself and my co-authors.